# Critical Factors for the Success of Rural Water Supply Services in Brazil

**Anna V. M. Machado \***[ID]**, João A. N. dos Santos, Norbertho da S. Quindeler**[ID] **and Lucas M. C. Alves**

Departamento de Desenho Técnico, Escola de Engenharia, Universidade Federal Fluminense (UFF), Rua Passos da Pátria, 156, Niterói, RJ, CEP 24210-240, Brazil; joaoalbertoneves@gmail.com (J.A.N.d.S.); norberthosq@id.uff.br (N.d.S.Q.); lucasmcarneiroalves@gmail.com (L.M.C.A.)

**\*** Correspondence: annav.machado@gmail.com; Tel.: +55-219-8112-3070

**Abstract:** The universalization of drinking water in rural communities poses a great challenge to developing countries, where rural areas often receive poor water service coverage and limited attention from authorities. This scenario is the current reality in Brazil. The community management model of rural water services has proven to be a noteworthy approach to ensure the continuity of water supply where private and public entities do not operate. However, its sustainability depends on several aspects. The authors of the current paper performed a thorough review of relevant publications in the rural sanitation field of study using the Preferred Reporting Items for Systematic Reviews (PRISMA) methodology, which enabled the creation of a list of essential factors capable to ensure the sustainability of Rural Water Supply Services (RWSS). Using the Nominal Group Technique with a selection of participants from a national conference held in Brazil in 2015, specialists hierarchized the factors, demonstrating their perception of the most important aspects necessary in RWSS throughout Brazil. Consequently, the authors noticed the necessity of a strong enabling environment, which recognizes small communities and their local services. Water quality control, post-construction support and the existence of a financial scheme were also pointed out as important aspects to ensure RWSS's sustainability.

**Keywords:** Brazil; drinking water; rural areas; sustainability

---

## 1. Introduction

Granting access to safe water for humanity has been a major challenge to all countries around the world. From 2000 to 2015, local, regional, and global efforts managed to increase the number of people with access to at least basic drinking water services to 89% of the entire population of the planet. This number reveals approximately 844 million people are still left behind with no access to any kind of safe water source [1], which contributes to decreased quality of life [2]. According to WHO and UNICEF [3] and Marks et al. [4], rural areas are one of the key challenges in this issue, especially those located in least developed countries. Rural settlements are usually located in remote areas, turning the expansion of existing water services or the creation of local water services into an unfeasible operation if best practices are not executed. Hence, these particular areas demand closer attention and increased efforts from decision-makers to guarantee the fulfilment of the Sustainable Development Goal 6, which consists of delivering drinking water for all by 2030 [1,5].

An important management model established in remote communities worldwide to facilitate access to drinking water is the community management model of water supply systems. In this model, community organizations are responsible for the operation and maintenance of water services, including the water treatment process, billing and maintenance procedures. There is no specific

framework for its initiation, since its sustainability requires adaptation to local contexts regarding economic, social and political aspects [6–8]. However, principles such as the promotion of community participation and the existence of a solid financial scheme represent common features of successful cases throughout the world. Calzada et al. [9] concluded the presence of Juntas Administradoras de Servicios de Saneamiento (JAAS), community-managed organizations responsible to manage water supply systems where local governments could not provide quality services, was critical for positive outcomes in Peru. Rautanen and White [10] had similar findings in Nepal, where community organizations had been able to efficiently maintain water services in some small towns where well-structured institutional and financial practices had been adopted. Meanwhile, Barde [11] revealed that water supply projects with active participation of users were able to grant access to safe water to a greater number of people when compared to governmental approaches in rural communities of Brazil. Thus, community management has a pivotal role in providing drinking water to communities where public and private interests are minimal.

Nevertheless, this management model has exhibited several weaknesses which have been explored in scientific literature for several decades. Some authors observed that the lack of institutional support and political planning are major contributors to systems' failures, as seen in Ghana [5] and Nigeria [12]. Insufficient financial resources and weak tariff schemes also have been recognized as leading causes of failure of rural water services [8,13–15]. Additionally, several studies identified maintenance issues as critical to the functionality of community-managed systems, such as the execution of insufficient preventive repairs [16], difficulties to obtain spare parts [12] and lack of technical support from external entities [17,18]. Therefore, community management models of water supply require further commitment from multiple stakeholders in order to achieve better results and benefit a larger number of people [19].

Through this work, we assessed the perceptions of Brazilian specialists with professional backgrounds in the rural water sector. Their practical experience provided a valuable interpretation of the daily difficulties of Rural Water Supply Services (RWSS), and allowed for a constructive assessment of the key elements associated with the success of community management. The Brazilian scenario regarding RWSS is currently in the imminence of major modifications as a result of the elaboration of a national plan (Plano Nacional de Saneamento Rural), which will be approached later in this work. Understanding the perception of specialists regarding the necessities of RWSS could provide significant insights and contributions during the elaboration phase of the national plan. This work used the Nominal Group Technique (NGT). The perceptions of specialists were collected through the application of a survey containing 30 factors based on relevant scientific publications on the rural water supply field of study. Some of the publications used in the current work and their relevant conclusions are exposed in the next section. The documents used to create the survey were a result of the application of the Preferred Reporting Items for Systematic Reviews (PRISMA), which will be described thoroughly in Section 2. From these factors, specialists were asked to name the six most important factors to determine which are the most critical for RWSS in Brazil. From their perspectives, it was possible to understand the elements deemed most important to address to assist in the development of more effective strategies and policies for the RWSS sector in Brazil.

*Critical Factors for Community Management Success in the Literature*

A significant number of publications have confirmed the need for constant post-construction support from external agencies regarding technical, managerial and financial aspects to ensure the long-term sustainability and functionality of RWSS [20–26]. In a study developed in Dominican Republic, Schweitzer and Mihelcic [27] found that the frequency of technical visits from external agencies responsible to offer support to rural service providers was positively related to communities where a strong financial scheme persisted. Moriarty et al. [28] also recognized the necessity of external support to ensure the successfulness of RWSS, which is aligned to the same conclusions from Smits et al. [29], Vásquez [30] and Hutchings et al. [8]. This external support must develop

capacities beyond the technical knowledge required to operate a water treatment plant, which is a recommendation stated in Rivas et al. [31]. The authors noticed a high number of failures in cases where only technical support was provided, ignoring managerial and governance aspects.

The requirement of strong financial schemes to adequately maintain RWSS is also a consensus among researchers [32–34], even though it is necessary to maintain water services at an affordable price level to ensure the capacity-to-pay of populations with varying economic levels. Vásquez [30] and Behnke et al. [14] support the development of financial configurations which allow users from low-income households to afford the water tariffs. They argue that the exchange of labor and other personal assets, such as crop yields, should be accepted as alternatives to capital contributions, allowing users to maintain their water connection when money is scarce. Nevertheless, the main practice adopted in financial schemes of RWSS relies on periodic monetary charges. In these cases, tariffs must be affordable to users and still be able to cover costs of operation, maintenance and refurbishment of systems [35]. The capability to refurbish the entire water supply structure using revenue from water tariffs is indicated as a desirable aspect of financial schemes of RWSS, otherwise capital infusion would be necessary, which is not a reality for most developing countries [31].

The capability of tariff schemes to maintain a RWSS adequately active depends strongly on community awareness on the necessity of the service and its acceptability by users. Many authors interpret these notions as users' willingness-to-pay for the service, an aspect which requires a thorough engagement from decision-makers to be built [36,37]. Promoting the participation of users during planning and implementation phases of RWSS has proven to be an effective strategy to increase their willingness-to-pay for the service as inhabitants usually develop a sense of ownership towards the water system, and are willing to contribute monetarily to maintain the service [10,13,38,39]. Consequently, service providers are able to decrease their dependence upon external financial subsidies.

To ensure continuous sustainability of RWSS, authors have argued that decision-makers should consider the enabling environment one of its most influential pillars. According to Amjad et al. [40], a solid enabling environment should be capable to aid rural service providers in offering sustainable and high-quality water services. This could be achieved through the development of regulations, policies, funding and monitoring programs and comprehensive frameworks which detail the roles and responsibilities of the stakeholders involved [41–44]. Mandara et al. [45] and Trémolet [35] argue that a well-defined set of regulations comprising economic, environmental and public health elements which are flexible to adapt to varying realities in a country is a key element to promote the sustainability of RWSS. Berhane [46] and Moriarty et al. [28] emphasize the importance of explicit frameworks, as the authors recognize that several countries already acknowledge community organizations as formal service providers; however, they fail to provide clear information regarding their actual roles. This was also a reality in Uganda, for example, according to Quin et al. [47]. The combination of these aspects allows community-based service providers to encounter favorable conditions to operate and manage their organizations and water services properly. Moreover, as stated by Marks et al. [48], Kayser et al. [49] and Cronk and Bartram [50], the establishment of a solid enabling environment depends on the availability of sufficient data obtained through monitoring programs able to illustrate the actual conditions of RWSS through multiple indicators, reaffirming the importance of developing monitoring mechanisms.

## 2. Materials and Methods

### 2.1. Nominal Group Technique (NGT) and Selection of Relevant Publications

The data used in this article was collected using the Nominal Group Technique (NGT) approach. The NGT is a decision-making tool widely applied in managerial researches as a strategy to accelerate the generation of decisions, requiring participants to individually write down ideas regarding the possible solutions to a determined issue without any influence from other participants. Afterwards, the ideas are exposed to a group of participants, who subsequently analyze them, determine those

with most significance to the problem and hierarchize them to facilitate decision-making processes [51]. When this methodology is able to assemble a group of participants from different backgrounds, it enables the generation of a diversified and, possibly, unconnected list of solutions to a specific issue, contributing to the development of varied strategies to problem-solving.

The authors of the current article performed the first stage of the NGT. Several key elements appointed as decisive in community management of RWSS were assembled from scientific literature into a list. The publications used in the current study were selected using the Preferred Reporting Items for Systematic Reviews (PRISMA) [52], using the "Google Scholar" search engine. The search was performed twice. Firstly, it was executed in June, 2015, considering publications from 2009 until the date of application of the survey. In June, 2019, another round of the search was executed to gather articles from more recent studies which could reaffirm the significance of the factors of the list. The same keywords were used in both rounds, which were "water supply", "sustainability", "rural areas", "rural water supply" and "community management". Only documents in English, Portuguese and Spanish were selected due to language restrictions of the authors. The first and second rounds resulted in 680 and 997 publications, respectively. For the second round, 34 duplicates were excluded, resulting in 963 publications for the title screening process, which led to 459 publications with relevant titles to the scope of the current work. The following step was based on a screening of the abstract of resulting publications to ensure their accordance to the RWSS theme and to assess the quality of the research methods each one utilized. As a result, 111 publications were considered for full-text analysis, being 53 the number of publications utilized to support the final list due to the fact that they propose factors which could increase RWSS's sustainability. In addition, the others were selected as they were related to situations and assessments similar to the Brazilian's scenario regarding RWSS.

The list consists of 30 factors involving economical, institutional, political, managerial and technical aspects in the RWSS field of study. Participants were asked to select six factors from the total amount, representing their opinion on which aspects should be prioritized in a decision-making processes to create the best outcomes in community management models of RWSSs in Brazil. The factors are represented in Table 1.

**Table 1.** Display of the 30 critical factors associated with successful cases of the community management model of rural water supply.

| No. | Factor |
| --- | --- |
| 1 | Existence of regulations regarding the establishment of charges to users and the maintenance of quality standards of the water delivered as a form of protection of the consumer |
| 2 | Establishment of adequate periodic charges capable of covering operational and maintenance costs |
| 3 | Establishment of adequate periodic charges capable of covering operational, maintenance and spare parts costs |
| 4 | Establishment of adequate periodic charges capable of covering costs of the life cycle of the equipment of the systems |
| 5 | Provision of sufficient subsidies to cover part of the costs associated to refurbishment of the system |
| 6 | Establishment of periodic charges according to the local context with fares aligned to user's affordability |
| 7 | Requirement of environmental licensing of the water catchment and treatment plant |
| 8 | Comply with water catchment limits determined in the environmental license |
| 9 | Accordance to water quality standards |
| 10 | External support for the execution of sophisticated water quality tests |
| 11 | Existence of environmental laws regarding water and sanitation which specify and recognize community management systems of water supply in rural communities as service providers |
| 12 | Existence of a favorable political environment for the creation of political and institutional marks, financial planning and the development of innovations at national level |

**Table 1.** *Cont.*

| No. | Factor |
|---|---|
| 13 | Existence of a favorable political environment for the creation of political and institutional marks, financial planning and the development of innovations at state level |
| 14 | Existence of local entities specialized in planning, contracting, regulation and support of local service providers |
| 15 | Existence of local service providers, community-managed or not, responsible to operate, manage and maintain local systems |
| 16 | Professionalization of community-managed services in communities where volunteer schemes prevail |
| 17 | Existence of monitoring indicators of system's functionality and sustainability |
| 18 | Existence of structured support systems for the post-construction phase to assist local communities and service providers |
| 19 | Execution of periodic capacity-building activities with local authorities to promote best practices in planning and management of water supply systems in rural communities |
| 20 | Execution of capacity-building and knowledge management activities at national level to promote successful cases achieved in the country |
| 21 | Formalization of community organizations and of their role as water service providers |
| 22 | Formalization of community organizations roles in water provision and their relationship with local governments |
| 23 | Existence of investments in capacity-building and technical support in the post-construction phase |
| 24 | Existence of a regional entity responsible to facilitate contact between local communities and national government to support community management systems regarding technical and management support provision |
| 25 | Existence of partnerships between local communities and local governments to develop community management systems regarding technical, institutional and operational aspects |
| 26 | Existence of partnerships between local communities and private entities to develop community management systems regarding technical, institutional and operational aspects |
| 27 | Adoption of appropriate technologies considering local aspects and feasibility according to user's capability to afford periodic charges |
| 28 | Strengthening of community participation and development of the sense of ownership of users |
| 29 | Solid institutional structure of the community organization, being legally recognized |
| 30 | Establishment of accountability as requirement for good governance |

## 2.2. Data Collection

The collection of data was executed during the 28th Brazilian Conference on Environmental and Sanitary Engineering, held in Rio de Janeiro, in 2015, a biannual event which unites professionals and researchers from a variety of fields in water and sanitary engineering. Considering the potential of the event to gather a significant amount of water specialists in one location, the authors were divided into several teams to approach conference attendees. The teams also counted with the aid of volunteers to conduct the surveys. The selection criterion of participants was based on a sample of conference attendees who were present in the conference on the day when sessions related to the theme "rural sanitation" were conducted. These sessions covered a variety of topics in the rural sanitation and water supply field of study, debating problems, good-practices, management and instructional arrangements, considering national, regional and local levels, regarding the water supply systems in rural communities in Brazil. Additionally, the selection considered attendees' availability and willingness to participate in the study during the course of the event. Participants were asked to fill a consent form after the interviewers exposed the purpose of the study and the methodology which

would be applied, allowing the usage of their responses in the current study if they were willing to continue.

Then, participants were asked to thoroughly analyze the list and to rank the six most important factors according to their perception and expertise. Specialists were requested to perform this procedure individually, without any type of external influence from colleagues or the research team. Moreover, the 30 factors were randomly inserted into the list, as specialists could be influenced to select those placed higher in the answer sheet. The research limitation is that the data collection took place in a technical meeting, at national level; thus, the applicants were mostly professionals and managers working at national and regional levels, besides academic researchers. Therefore, a low number (*n* = 7) of survey attendees represented local communities.

*2.3. Data Analysis*

The analysis of the responses after completion of all surveys was based on two steps. First, each critical factor received a score according to their position in the hierarchy created by each participant. The scores ranged from 1 to 6, with the first most important critical factor receiving 6 points, which is the maximum score, and the sixth most important critical factor receiving 1 point. After the analysis of each survey, the total score of each factor was computed. For the second step, the authors calculated the number of appearances of each factor in the total amount of answer sheets. Subsequently, a percentage of appearance was determined for each factor, which was then used to multiply the total score computed for the critical factor, resulting in the Relative Weight (RW), calculated using the equation *RW = total score x (number of surveys in which the factor appears)/(total number of surveys)*. This method was developed to consider the influences of the scores and percentages of appearances equally. In case a factor received a high score but was considered by a low number of participants, it would be equally recognized and ranked with a factor which received lower scores but was present in most of the answers.

## 3. Results

The study was able to collect a total of 88 complete surveys from either professionals and researchers from the rural water supply field. The background experience of respondents in this field of study varied. Nearly half of interviewees expressed more than 5 years of practical knowledge regarding RWSS (48.86%). From the total, 15 interviewees had completed a Doctoral Degree (17.05%), 21 had completed a Master's Degree (23.86%), 31 had pursued other types of graduate certificates (35.23%), and 9 had completed a Bachelor's Degree (10.23%). Most participants (*n* = 46) worked in institutions operating at national level (52.27%), while 19.32% of the interviewees (*n* = 17) worked at state level, 10.23% (*n* = 9) worked at municipal level, and 7.95% (*n* = 7) worked at local communities' level. The remaining interviewees worked at regional organizations. Furthermore, this research was able to collect contributions from professionals covering a wide extension of the Brazilian territory, as they represented 20 of a total of 26 Brazilian states (76.92%). This study could not collect contributions from specialists working at international organizations.

The critical factors which received the highest RW were Factors 12 (RW = 60.45—Existence of a favorable political environment for the creation of political and institutional marks, financial planning and the development of innovations at national level), 9 (RW = 48.00—Accordance to water quality standards), 14 (RW = 47.33—Existence of local entities specialized in planning, contracting, regulation and support of local service providers), 6 (RW = 43.64—Establishment of periodic charges according to the local context with fares aligned to user's affordability), 18 (RW = 27.00—Existence of structured support systems for the post-construction phase to assist local communities and service providers) and 15 (RW = 18.64—Existence of local service providers, community-managed or not, responsible to operate, manage and maintain local systems), and the hierarchy of the entire list of factors is demonstrated in Figure 1. The six factors with higher RWs were also the most commonly identified in the surveys. Factor 12 stands out as the most significant element which should be addressed according

to specialists. Factors 9, 14 and 6 appear as relevant elements when compared to the others. The RWs of factors 18 and 15 presented lower differences when comparing to the remaining part of the list; however, the last two factors also stand out as more relevant considering the low RWs of the majority of the remaining factors.

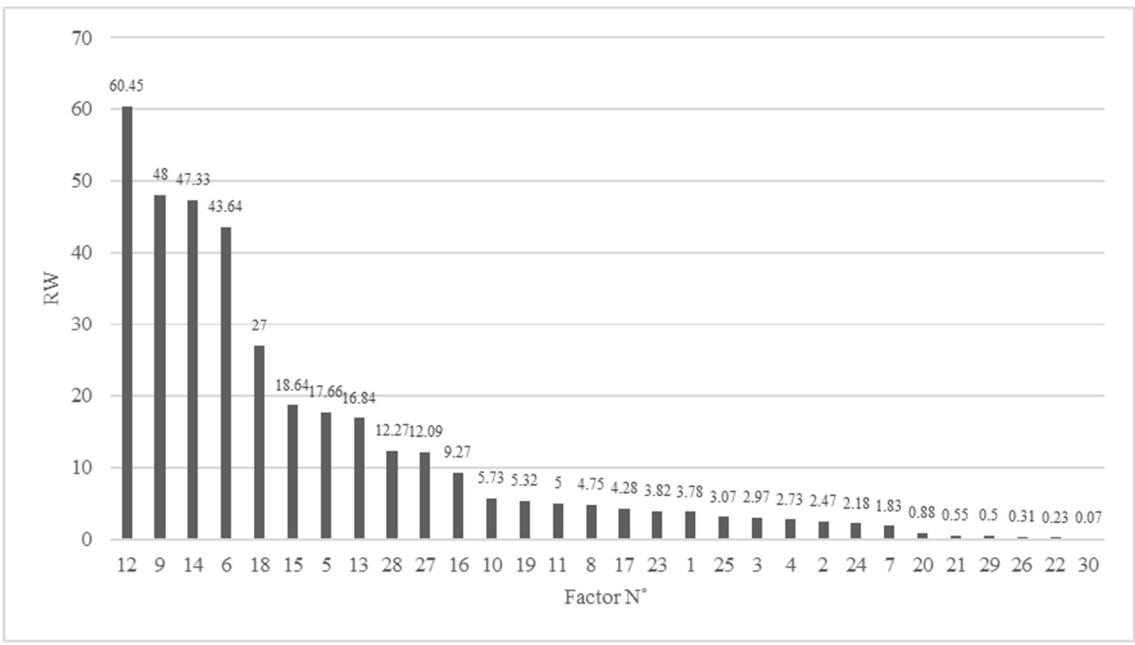

**Figure 1.** Representation of the critical factors ordered by relative weight (RW).

## 4. Discussion

### 4.1. Factor 12: Existence of a Favorable Political Environment for the Creation of Political and Institutional Marks, Financial Planning and the Development of Innovations at National Level

Recent data regarding rural areas in Brazil are scarce, the last national data collection having been performed in 2016. According to Instituto Brasileiro de Geografia e Estatística (IBGE) [53], 15.64% (29,830,007 inhabitants) of Brazilian population lived in rural areas of the country during the time of the last national census, which represents a significant number of people, considering it surpasses the population of several European countries. Therefore, it is evident that Brazil's decision-makers possess a great responsibility towards the quality of life and public health of an expansive number of people. Considering data collected by IBGE [54], only 34.50% of rural households could rely on safe water connections by the time of the study, while the number of urban households connected to reliable water services reached 93.90%. This unbalanced scenario is a consequence of several decades of negligence from federal, state and municipal governments towards the improvement of the quality of life of rural settlers. This is recognized by professionals of this sector according to the results of the current work. The strong necessity of the creation of a national enabling environment which favors the existence of community organizations and their service delivery model is well-represented by the position of this factor in the hierarchy created by the current study.

Currently, the federal law which allocates the responsibilities regarding drinking water delivery in Brazil is Law No. 11.445/2007, which determines municipalities and the Federal District as accountable for water services, being also their responsibility to create municipal plans used to determine guidelines to sanitation, water and solid waste collection services. Municipalities are able to allocate these services to third parties through the establishment of a formal contract [55]. Therefore, water services are operated by private, public or mixed capital entities in urban areas of Brazil. These entities usually do not expand their services to rural areas due to the unfeasibility to expand water services to regions where costs of operation and maintenance would likely overcome revenue. In these rural and isolated

areas, Law No. 11.445/2007 allows the concession of water services without the requirement of a formal contract to rural settlers organized in associations [55], referring directly to the community management model. Besides this mentioning, no other evidence of the model exists in the Brazilian federal law, with no national guideline or framework to guide local service providers and other stakeholders in the rural water supply sector.

Municipal plans rarely mention populated areas beyond the urban limits of the city, neglecting or poorly approaching isolated and rural communities [56]. The Plano Nacional de Saneamento Básico (Basic Sanitation National Plan), a federal document created by Ministério das Cidades (Cities Ministry), was developed as a national framework to guide authorities from national, state and local levels in the provision of quality water and sanitation services [57]; however, even though the rural water supply sector is mentioned, the document provides a brief explanation of the challenges involved in this sector and mentions generalized strategies to overcome them. It is evident these aspects pose a major threat to the sustainability of community-managed water services as specific roles and responsibilities are not adequately allocated and could provoke the breakdown of rural water services due to low external assistance and involvement. The existence of a national framework to guide stakeholders in the pursuit of better rural water services is a necessity highlighted by Moriarty et al. [28], who support not only its existence, but also its comprehensiveness to avoid possible misinterpretations by the entities involved in the field, which could possibly hinder further advancements. According to the authors, several developing countries possess frameworks regarding drinking water delivery in rural areas; however, they are usually not explicit and fail to be effective, which is also the reality in Brazil. The Brazilian federal government has been developing the Plano Nacional de Saneamento Rural (National Plan of Rural Sanitation) since 2013, which is a collaborative initiative combining governmental agencies and several federal universities of the country in the chase for the establishment of national guidelines to improve service conditions and quality of life of rural areas. This document must allocate roles and responsibilities explicitly to ensure stakeholders engage adequately.

### 4.2. Factor 9: Accordance to Water Quality Standards

Accordance to water quality standards was also pointed to as a major aspect of sustainable community-managed rural water supply services. This reveals that specialists are aware of the potential threats caused by the delivery of a low-quality service to these communities. The delivery of drinking water with no risks to human health is one of the main drivers of the international pursuit of water universalization, minimizing the possibility of occurrence of water-related diseases in areas where they were once a great contributor to reduced quality of life. Once treated water becomes a reality in these communities, maintaining its quality above national standards becomes a necessity to maintain users' conviction that their financial contributions towards the service are being adequately deployed. Poor maintenance of water quality standards and the delivery of water with minimal to no reduction of risks to human health have the potential to discourage users to pay for the service. The willingness-to-pay for RWSS has proven to be a fundamental factor for the sustainability of several cases presented in the literature, as seen in Tigabu et al. [13], Kelly et al. [39] and Rautanen and White [10]. Hence, ensuring that local service providers are capable to deliver water which respects national standards of quality control consists of an important element to consider during the implementation and the entire life cycle of a RWSS.

### 4.3. Factor 14: Existence of Local Entities Specialized in Planning, Contracting, Regulation and Support of Local Service Providers

The existence of local entities specialized in the provision of managerial and institutional support to service providers during the entire life cycle of a RWSS was also an aspect highly considered by specialists interviewed in the current work, considering its third position in the final hierarchy. These entities present variations in their institutional configuration according to local contexts. As stated by Lockwood and Smits [58], these local entities normally occur in the form of service authorities,

such as municipalities or districts, being accountable to provide assistance to local service providers in planning, regulation and oversight procedures. In Brazil, there is a lack of external support for these aspects from local governments as consequence of a variety of internal problems faced by rural districts in the country, as the lack of commitment of local politicians after local elections, the embezzlement of public funds, and the low capacity of local staff [11,59]. These issues usually hinder the professionalization of community organizations, preventing the improvement of the quality of water services.

The inexistence of managerial and institutional support from local entities has led community organizations from some areas to create information networks as a form of self-assistance between water committees, commonly called associations. The associations of community organizations are a great response for the lack of external support, presenting several success cases where its presence has proven to generate benefits, such as constant technical, managerial and institutional support, increased recognition in decision-making processes, and higher levels of service, as seen in Dupuits and Bernal [60] and Machado et al. [61]. Machado et al. [61] presented some cases from Latin America, including the "Sistema Integrado de Saneamento Rural" (SISAR), which is an initiative from community organizations from the state of Ceará, Brazil, currently composed by representatives from water committees, and state and city councils. Its creation has enabled high levels of service in rural communities of Ceará through the provision of continuous support from the implementation to the post-construction phases.

### 4.4. Factor 6: Establishment of Periodic Charges According to the Local Context with Fares Aligned to User's Affordability

The establishment of periodic charges according to the affordability of users was also regarded as an important trigger to the sustainability of RWSS. This element poses a significant challenge to the community management model due to the fact that rural households often fail to possess sufficient capital to contribute effectively to maintain the service, especially in developing countries. According to Economic Commission for Latin America and the Caribbean (ECLAC) [62], 46.4% of rural inhabitants were facing poverty while 20.4% were below the extreme poverty line in 2017, illustrating an increase from previous years. Therefore, the usual economic conditions of rural settlers often impede the continuity of capital contributions from users to sustain the operation and maintenance of RWSS. These cases demand the establishment of charges according to the context in which the community is inserted. While there are communities where households are able to disburse the necessary amount of money on the periodicity determined by the service provider, a number of rural households must allocate their expanses towards other basic necessities, such as nutrition. Hence, a context-oriented definition of the best tariff scheme must be prioritized, as seen in the case presented by Behnke et al. [14] in some African countries, where the exchange of labor and personal assets for the access to drinking water was utilized to guarantee the delivery of potable water to inhabitants of varied income classes.

### 4.5. Factor 18: Existence of Structured Support Systems for the Post-Construction Phase to Assist Local Communities and Service Providers

Specialists from the current study also recognized the importance of post-construction support to service providers in order to maintain high levels of service. This element has been extensively approached through scientific publications, being one of the main factors associated to the successfulness of community-managed RWSS [8]. Post-construction support requires a high level of commitment from specialized entities, either public or private, which should maintain constant contact with service providers and provide a minimum amount of support to guarantee actual improvements to the RWSS, as stated by Moriarty et al. [28]. An interesting method to reduce community organizations' dependency on external support exists in some places in Ghana, as presented by Opare [63], where agencies from the district level provide technical support during the early phase of operation of the water service. During this period, the support provider gradually withdraws from the community to

ensure that the water committee is capable of running the RWSS with minimum interference from outside sources. Although the provision of external support must be reduced, it should not cease. This support provision must be a responsibility of agencies with the required technical skills and knowledge to effectively intervene in the RWSS without jeopardizing its initial conditions. This is a strong recommendation delivered by Smits et al. [29], whose study noticed that municipalities with low levels of professionalization were decreasing the performance of several RWSS in Colombia. Therefore, it is noticeable that the provision of external support to service providers is an element which possess several particularities that should be considered, otherwise its effectiveness could be greatly reduced.

*4.6. Factor 15: Existence of Local Service Providers, Community-Managed or Not, Responsible to Operate, Manage and Maintain Local Systems*

Lastly, specialists considered the necessity of the presence of a service provider to be the sixth critical factor associated to the success of RWSS. The service provider, either private, public, community-managed or any other type, is responsible to perform O&M activities routinely, being in constant contact with the water service and its daily challenges. The allocation of this factor between the six critical factors demonstrates a recent acknowledgement regarding the necessity to focus researches, initiatives and investments on the improvement of the performance of service providers, in contrast with the previous global trend to apply efforts towards almost exclusively hardware construction [28]. This trend was highly motivated by the Millennium Development Goals (MDGs) era, which defined as one of its international targets to halve the number of people without proper access to water resources. Even though the MDGs delivered great improvements to the quality of life of a significant number of people [64], their incapacity to address the safety and sustainability of services rose as a significant issue to tackle. Currently, the Sustainable Development Goal 6 poses a global target which aims to the universalization of safe drinking water for all, proposing several strategies to ensure the effectiveness and long-term reliability of water services throughout the world, as community participation and international cooperation [65]. Therefore, the SDGs aim beyond their former counterparts, addressing the necessity of efficient management systems operated by specialized service providers to generate the best outcomes.

## 5. Conclusions

The delivery of safe drinking water to rural and isolated communities poses a significant challenge to developing countries. In the case of Brazil, this challenge increases as a possible consequence of the large dimension of its national territory and the dispersion between communities in rural zones. Specialists of the rural water supply field were asked about their perception regarding the critical factors associated with the success of RWSS. The results demonstrate that Brazilian specialists consider a wide variety of aspects as essential to the promotion of high-quality services, such as the development of a national enabling environment, the establishment of adequate tariff schemes, the promotion of monitoring programs of water quality standards, the delivery of technical and institutional support, and the support to the enhancement of performance of service providers. Therefore, the aspects deemed as critical to the success of RWSS encompass political, economic, technical, managerial, and institutional elements, demonstrating the complexity of the delivery of water services in rural areas. It is noticeable that initiatives towards the universalization of drinking water access have expanded in Brazil, considering the current stage of development of the National Plan of Rural Sanitation (PNSR), and the existence of entities such as SISAR to support rural communities in the maintenance and daily operation of RWSS. The findings of this study reveal that specialists comprehend the complex scenario in which RWSS operate, and indicate that future decision-making processes in the Brazilian rural water supply sector could result in strategies which contemplate the extensive variety of aspects which influence the field. It is important to notice that a low number of specialists from the community level were present in the final number of participants of the current study. Their presence in future studies should be pursued as they can contribute with practical knowledge and experience, since they

are constantly confronting the challenges of RWSS. Nevertheless, the consideration of the six most critical factors recognized in the current work to the development of initiatives such as the PNSR should contribute to their effectiveness in improving the quality of life of a significant number of rural inhabitants.

**Author Contributions:** Conceptualization, A.V.M.M. and J.A.N.d.S.; methodology, L.M.C.A. and N.d.S.Q.; formal analysis, A.V.M.M., L.M.C.A and N.d.S.Q.; investigation, A.V.M.M., L.M.C.A. and N.d.S.Q.; writing—original draft preparation, A.V.M.M.; writing—review and editing, L.M.C.A. and N.d.S.Q.; visualization, L.M.C.A. and N.d.S.Q.; supervision, A.V.M.M.; project administration, A.V.M.M.

**Funding:** This research received no external funding.

**Conflicts of Interest:** The authors declare no conflict of interest.

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
