# Peer review of "Critical Factors for the Success of Rural Water Supply Services in Brazil"

_water, doi:10.3390/w11102180_

Round 1

Reviewer 1 Report

This paper is based on an interesting effort to gain an understanding from a captive 'expert field'.    It may be useful to provide the bibliography of the papers reviewed, short listed and selected to develop the question bank as an appendix to the paper so the sources of the information is transparent to the reader (e.g. what languages, journals, etc.).   There is also a considerable volume of grey literature that may have added to this effort, and for those who may wish to widen such a study, it would be helpful to understand the basic body of work used.

The paper should also present how the selection of interviewee met with ethical standards.  Were there any biases due to language choices of the publications or those completing questionnaires? 

The interpretation should be amended to acknowledge the scope and range of expertise, for instance of the 88 persons interviewed, how many were based out-with Brazil?   The respondents were to some degree self selected as they were willing to be involved.    Do the authors have some ability to place a confidence interval on the interpretation value for (a) Brazil and (b) outside Brazil?   It would strengthen the discussion and conclusions if some indication of confidence was provided.

My main suggestion for enhancing the manuscript would be to make sure that the results, while appearing to be valid for Brazil in this context, may require additional reasoning why the results are valid Globally.   This would make an average paper stronger.

Reviewer 2 Report

Overall, the paper deals with a topic of interest, particularly for the context of Brazil.  However, additional specificity is needed with respect to your methods and rationale for decisions that you made in the methods, in regards to your participants and their expertise, and your analysis.  If possible, this study could benefit from details on the systematic literature review and NGT in Supplementary Information, with additional results presented in SI (for instance, ratings by respondent type).  Additionally, your limitations need to be noted.  Finally, I believe the discussion could benefit from subtitles (if permitted) to orient the readers around the themes and how they relate to your results. 

The paper requires editing (e.g., specially versus especially in abstract, etc.)

Abstract- You should get to primary point (critical factors for successful RWS supply) and the method faster. The method (data collection and analysis) are unclear in the abstract as it just notes analysis of specialist knowledge and then a survey outcome.   Providing more of the results and what they mean is also needed.  For instance, the enabling environment is comprised of many things.   How will these findings facilitate decision-making?

1.0 Introduction

You need to indicate why Brazil, with contextual details that point to the need for the study and this context. Edit: What does “will be approached lately in this work” mean? Later? Some indication of this plan is needed upfront to orient the reader. More is needed in regards to how your literature review was conducted to determine the 30 factors under investigation (I recommend mentioning PRISMA in the Intro so people know that it is systematic.

2.0

Clarify the decision that you are analyzing using NGT. Provide the date range (indicate end dates of the literature review. Indicate why you selected the keyword that you did I’m unfamiliar with the options in this journal, but having Supplementary Information available for the PISMA lit review process and the NGT would be helpful if allowed. Was the lit review focused on similar contexts to Brazil?

2.2 

Did you collect information on the respondents to understand their expertise in community managed RWS services if they were randomly selected? Indicate how the score of each factor was computed. What was happening within the conference that may have contributed to answers? For instance, were there presentations occurring on RWS services before or during the survey?

3.0

Rather than indicate Factor 12, indicate what this is, at least in the writing. A deeper dive on items that appeared more than 10 on your scale of RW (still unclear how the total score is calculated) would be helpful. Another SI indicating additional information on the scores received for each category would also help, particularly broken down by type of respondent/respondent characteristics.

4.0      

I think it would help to discuss the primary results (top RW scores) by subtitle to help the readers.

Every study has limitations, what are yours?

Reviewer 3 Report

The paper is very interesting and it is well-written and presented.

- However the factors identified from the methodology used should be presented by name and not only by number.

The authors selected the questionnaires from experts in a national conference. This is not random sampling. How do the authors cope with the bias in the experts' selection?  What about the reliability of the answers? How the authors ensured that the answers are reliable?

Round 2

Reviewer 1 Report

The amendment's to the paper provides a transparent framework on which the results are interpreted and the context of the replies from the interviewees.  

Author Response

Dear Reviewer 1,

We appreciate your response regarding the latest version of our work. Hope our study is easily comprehensible and is able to be relevant in this field of study.

Sincerely,

Anna Virgínia.

Reviewer 3 Report

The authors provided an improved revised version of their manuscript.

As their research has been done using a rather not random sample, it is necessary to mention this face as a limitation in their research. I suggest that the authors should clearly state the research's limitations in conclusions.

Author Response

Dear Reviewer 3,

We made the last few modifications suggested. Our limitations are now presented in the conclusions so the reader can consider them when reading our final thoughts. We appreciate your response and help to improve our manuscript and turn it into a relevant contribution.

Sincerely,

Anna Virgínia.